# Rethinking the compositionality of point clouds through regularization in the hyperbolic space

**Antonio Montanaro**∗
Politecnico di Torino, Italy
antonio.montanaro@polito.it

**Diego Valsesia**
Politecnico di Torino, Italy
diego.valsesia@polito.it

**Enrico Magli**
Politecnico di Torino, Italy
enrico.magli@polito.it

## Abstract

Point clouds of 3D objects exhibit an inherent compositional nature where simple parts can be assembled into progressively more complex shapes to form whole objects. Explicitly capturing such part-whole hierarchy is a long-sought objective in order to build effective models, but its tree-like nature has made the task elusive. In this paper, we propose to embed the features of a point cloud classifier into the hyperbolic space and explicitly regularize the space to account for the part-whole hierarchy. The hyperbolic space is the only space that can successfully embed the tree-like nature of the hierarchy. This leads to substantial improvements in the performance of state-of-art supervised models for point cloud classification.

## 1   Introduction

Is the whole more than the sum of its parts? While philosophers have been debating such deep question since the time of Aristotle, we can certainly say that understanding and capturing the relationship between parts as constituents of whole complex structures is of paramount importance in building models of reality. In this paper, we turn our attention to the compositional nature of 3D objects, represented as point clouds, where simple parts can be assembled to form progressively more complex shapes. Indeed, the complex geometry of an object can be better understood by unraveling the implicit hierarchy of its parts. Such hierarchy can be intuitively captured by a tree where nodes close to the root represent basic universal shapes, which become progressively more complex as we approach the whole-object leaves. Transforming an object into another requires swapping parts by traversing the tree up to a common ancestor part. It is thus clear that a model extracting features, that claim to capture the nature of 3D objects, needs to incorporate such hierarchy.

In the last years, point cloud processing methods have tried to devise methods to extract complex geometric information from points and neighborhoods. Architectures like graph neural networks [1] compose the features extracted by local receptive fields, with sophisticated geometric priors [2] exploiting locality and self-similarity, while a different school of thought argues that simple architectures, such as PointMLP [3] and SimpleView [4], with limited geometric priors are nevertherless very effective. It thus raises a question whether prior knowledge about the data is being exploited effectively.

In this sense, works such as PointGLR [5], Info3D [6] and DCGLR [7] recognized the need to reason about local and global interactions in the feature extraction process. In particular, their claim is that

---

∗Code of the project: https://github.com/diegovalsesia/HyCoRe

36th Conference on Neural Information Processing Systems (NeurIPS 2022).

maximizing the mutual information between parts and whole objects leads to understanding of local and global relations. Although these methods present compelling results for unsupervised feature extraction, they still fall short of providing significant improvements when finetuned with supervision.

In our work, we argue that those methods do not fulfill their promise of capturing the part-whole relationship because they are unable to represent the tree-like nature of the compositional hierarchy. Indeed, their fundamental weakness lies in the use of spaces that are either flat (Euclidean) or with positive curvature (spherical). However, it is known that only spaces with negative curvature (hyperbolic) are able to embed tree structures with low distortion [8]. This is due to the fact that the volume of the Euclidean space grows only as a power of its radius rather than exponentially, limiting the representation capacity of tree-like data with an exponential number of leaves. This unique characteristic has inspired many researchers to represent hierarchical relations in many domains, from natural language processing [9],[10] to computer vision [11] ,[12]. However, the use of such principles for point clouds and 3D data is still unexplored.

The main contributions of this paper lie in the following aspects:

- we propose a novel regularizer to supervised training of point cloud classification models that promotes the part-whole hierarchy of compositionality in the hyperbolic space;

- this regularizer can be applied to any state-of-art architecture with a simple modification of its head to perform classification with hyperbolic layers in the regularized space, coupled with Riemannian optimization [13];

- we observe a significant improvement in the performance of a number of popular architectures, including state-of-the-art techniques, surpassing the currently known best results on two different datasets;

- we are the first to experimentally observe the desired part-whole hierarchy, by noticing that the geodesics in hyperbolic space between whole objects pass through common part ancestors.

## 2 Related work

**Point Cloud Analysis** Point cloud data are sets of multiple points and, in recent years, several deep neural networks have been studied to process them. Early works adapted models for images through 2D projections [14], [15]. Later, PointNet [16] established new models working directly on the raw set of 3D coordinates by exploiting shared architectures invariant to points permutation. Originally, PointNet independently processed individual points through a shared MLP. To improve performance, PointNet++ [17] exploited spatial correlation by using a hierarchical feature learning paradigm. Other methods [18], [19], [20], treat point clouds as a graph and exploit operators defined over irregular sets to capture relations among points and their neighbors at different resolutions. This is the case of DGCNN [21], where the EdgeConv graph convolution operation aggregates features supported on neighborhoods as defined by a nearest neighbor graph dynamically computed in the feature space. Recently, PointMLP [3] revisits PointNet++ to include the concept of residual connections. Through this simple model, the authors show that sophisticated geometric models are not essential to obtain state-of-the-art performance.

**Part Compositionality** Successfully capturing the semantics of 3D objects represented as point clouds requires to learn interactions between local and global information, and, in particular, the compositional nature of 3D objects as constructed from local parts. Indeed, some works have focused on capturing global-local reasoning in point cloud processing. One of the first and most representative works is PointGLR [5]. In this work, the authors map local features at different levels within the network to a common hypersphere where the global features embedding is made close to such local embeddings. This is the first approach towards modeling the similarities of parts (local features) and whole objects (global features). The use of a hypersphere as embedding space for similarity promotion traces its roots in metric learning works for face recognition [22]. In addition to the global-local embedding, PointGLR added two other pretext tasks, namely normal estimation and self-reconstruction, to further promote learning of highly discriminative features. Our work significantly differs from PointGLR in multiple ways: i) a positive curvature manifold such as the hypersphere is unable to accurately embed hierarchies (tree-like structures), hence our adoption of the hyperbolic space; ii) we actively promote a continuous embedding of part-whole hierarchies by

penalizing the hyperbolic norm of parts proportionally to their number of points (a proxy for part complexity); iii) we move the classification head of the model to the hyperbolic space to exploit our regularized geometry. A further limitation of PointGLR is the implicit assumption of a model generating progressive hierarchies (e.g. via expanding receptive fields) in the intermediate layers. In contrast, our work can be readily adopted by any state-of-the-art model with just a replacement of the final layers. Other works revisit the global-local relations using maximization of mutual information between different views [6], clustering and contrastive learning [23], distillation with constrast [7], self-similarity and contrastive learning with hard negative samples [24]. Although most of these works include the contrastive strategy, they differ in the way they contrast the positive and negative samples and in the details of the self-supervision procedures, e.g., contrastive loss and point cloud augmentations. We also notice that most these works focus on unsupervised learning, and, while they show that the features learned in this manner are highly discriminative, they are also mostly unable to improve upon state-of-the-art supervised methods when finetuned with full supervision. These approaches differ from the one followed in this paper, where we focus on regularization of a fully supervised method, and we show improvements upon the supervised baselines that do not adopt our regularizer.

**Hyperbolic Learning** The intuition that the hyperbolic space is crucial to embed hierarchical structures comes from the work of Sarkar [8] who proved that trees can be embedded in the hyperbolic space with arbitrarily low distortion. This inspired several works which investigated how various frameworks of representation learning can be reformulated in non-Euclidean manifolds. In particular, [9] [13] and [10] were some of the first works to explore hyperbolic representation learning by introducing Riemannian adaptive optimization, Poincarè embeddings and hyperbolic neural networks for natural language processing. The new mathematical formalism introduced by Ganea et al. [10] was decisive to demonstrate the effectiveness of hyperbolic variants of neural network layers compared to the Euclidean counterparts. Generalizations to other data, such as images [25] and graphs [26] with the corresponding hyperbolic variants of the main operations like graph convolution [26] and gyroplane convolution [12] have also been studied. In the context of unsupervised learning, new objectives in the hyperbolic space force the models to include the implicit hierarchical structure of the data leading to a better clustering in the embedding space [12], [11]. To the best of our knowledge, no work has yet focused on hyperbolic representations for point clouds. Indeed, 3D objects present an intrinsic hierarchy where whole objects are made by parts of different size. While the smallest parts may be shared across different object classes, the larger the parts the more class-specific they become. This consistently fits with the structure of a tree where simple fundamental parts are shared ancestors of complex objects and hence we show how the hyperbolic space can fruitfully capture this data prior.

# 3 Method

In this section we present our proposed method, named HyCoRe (Hyperbolic Compositional Regularizer). An overview is presented in Fig. 1. At a high level, HyCoRe enhances any state-of-the-art neural network model for point cloud classification by 1) replacing its last layers with layers performing transformations in the hyperbolic space (see Sec. 3.2), and 2) regularizing the classification loss to induce a desirable configuration of the hyperbolic feature space where embeddings of parts both follow a hierarchy and cluster according to class labels.

## 3.1 Compositional Hierarchy in 3D Point Clouds

The objective of HyCoRe is to regularize the feature space produced by a neural network so that it captures the compositional structure of the 3D point cloud at different levels. In particular, we notice that there exists a hierarchy where small parts (e.g., simple structures like disks, squares, triangles) composed of few points are universal ancestors to more complex shapes included in many different objects. As these structures are composed into more complex parts with more points, they progressively become more specific to an object or class. This hierarchy can be mathematically represented by a tree, as depicted in Fig.2 where a simple cylinder can be the ancestor of both pieces of a chair or a table. While the leaves in the tree are whole objects, thus belonging to a specific class, their ancestors are progressively more universal the higher up in the hierarchy they sit.

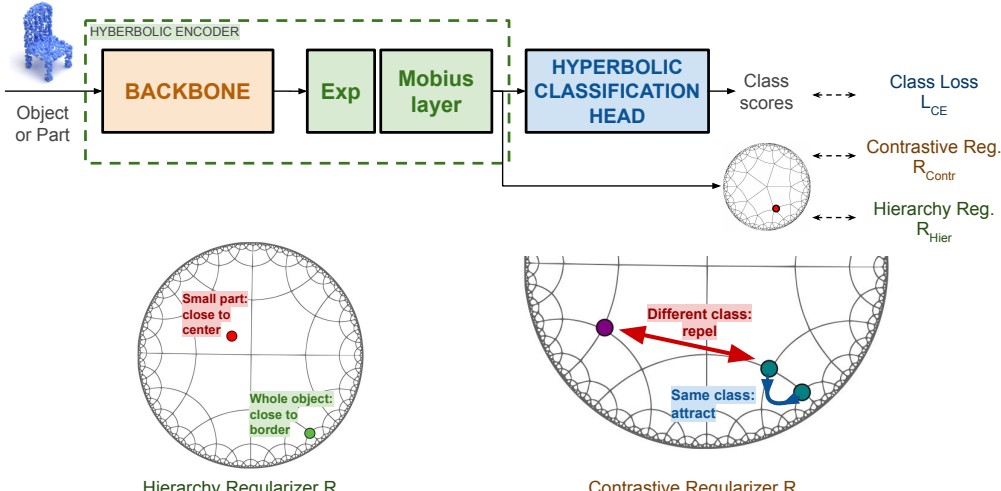

Figure 1: HyCoRe overview. A point cloud classification model is regularized by promoting the feature space to include compositional information. Hierarchy regularizer: simple parts should be mapped closer to the center of the Poincarè disk (common ancestors of whole objects). Contrastive regularizer: parts of the same class should be embedded closer than parts of other classes.

At this point, it is important noting that the graph distance between leaves is determined by the shortest path passing through the first common ancestor for objects in the same or similar classes, while objects from significantly dissimilar classes have the shortest path passing through the root of the hierarchy. In order to ensure that we can embed this tree structure in a feature space, we need a space that preserves the geometrical properties of trees and especially the graph distance. In particular, the embedding space must be able to accommodate the exponential volume growth of a tree along its radius. A classic result by Sarkar [8] showed that flat Euclidean space does not provide this, leading to high errors when embedding trees, even in high dimensions. On the contrary, the hyperbolic space, a Riemannian manifold with negative curvature, does support exponentially increasing volumes and can embed trees with arbitrarily low distortion. Indeed, the geodesic (shortest path) between two points in this space does pass through points closer to the origin, mimicking the behavior of distance defined over a tree.

In particular, we will focus on the Poincarè ball model of hyperbolic space. Since hyperbolic space is a non-Euclidean manifold, it cannot benefit from conventional vector representations and linear algebra. As a consequence, classical neural networks cannot operate in such a space. However, we will use extensions [10] of classic layers defined through the concept of gyrovector spaces.

### 3.2 Hyperbolic Space and Neural Networks

The hyperbolic space is a Riemannian manifold with constant negative curvature. The curvature determines the metric of a space by the following formula:

$$\mathbf{g}_R = (\lambda_x^c)^2 \mathbf{g}_E = \frac{2}{1 + c\|\mathbf{x}\|^2} \mathbf{g}_E \tag{1}$$

where $\mathbf{g}_R$ is the metric tensor of a generic Riemannian manifold, $\lambda_x^c$ is the conformal factor that depends on the curvature $c$ and on the point $\mathbf{x}$ on which is calculated, and $\mathbf{g}_E$ is the metric tensor of the Euclidean space $\mathbb{R}^n$, i.e., the identity tensor $\mathbf{I}_n$. Note how the metric depends on the coordinates (through $\|\mathbf{x}\|$) for $c \neq 0$, and how $c = 0$ yields $\mathbf{g}_R = 2\mathbf{g}_E$, i.e., the Euclidean space is a flat Riemannian manifold with zero curvature. Spaces with $c > 0$ are spherical, and with $c < 0$ hyperbolic.

The Poincarè Ball in $n$ dimensions $\mathbb{D}^n$ is a hyperbolic space with $c = -1$, and it is isometric to other models such as the Lorentz model. The distance and norm are defined as:

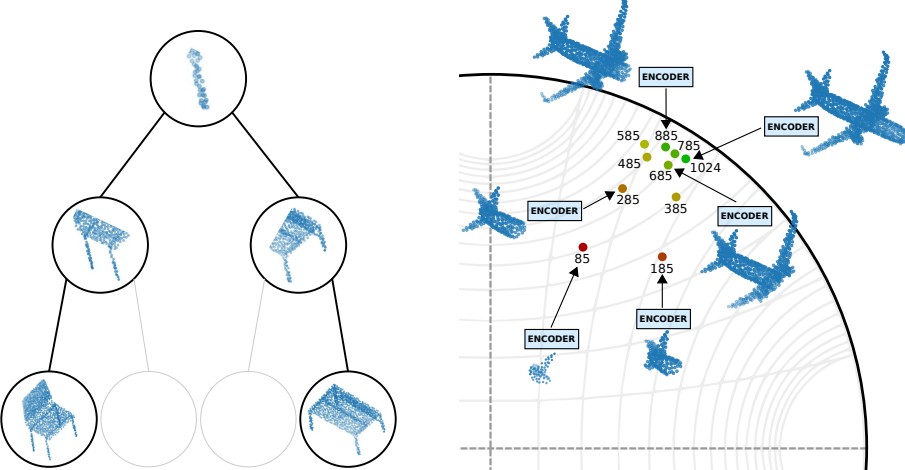

Figure 2: 3D objects possess inherent hierarchies due to their nature as compositions of small parts. The hyperbolic space can embed trees and hierarchical structures with lower distortions than the Euclidean space. The number of points in the embedded part point cloud is highlighted in figure. Embeddings shown are experimental results projected to 2D Poincarè disk with hyperbolic UMAP.

$$d_{\mathbb{D}}(\mathbf{x}, \mathbf{y}) = \cosh^{-1}\left(1 + 2\frac{\|\mathbf{x} - \mathbf{y}\|^2}{(1 - \|\mathbf{x}\|^2)(1 - \|\mathbf{y}\|^2)}\right)\right), \qquad \|\mathbf{x}\|_{\mathbb{D}} = 2\tanh^{-1}\left(\|\mathbf{x}\|\right) \qquad (2)$$

Since the Poincarè Ball is a Riemannian manifold, for each point $\mathbf{x} \in \mathbb{D}^n$ we can define a logarithmic map $\log_{\mathbf{x}} : \mathbb{D}^n \to T_{\mathbf{x}}\mathbb{D}^n$ that maps points from the Poincarè Ball to the corresponding tangent space $T_{\mathbf{x}}\mathbb{D}^n \in \mathbb{R}^n$, and an exponential map $\exp_{\mathbf{x}} : T_{\mathbf{x}}\mathbb{D}^n \to \mathbb{D}^n$ that does the opposite. These operations [10] are fundamental to move from one space to the other and viceversa.

The formalism to generalize tensor operations in the hyperbolic space is called the gyrovector space, where addition, scalar multiplication, vector-matrix multiplication and other operations are redefined as Möbius operations and work in Riemannian manifolds with curvature $c$. These become the basic blocks of the hyperbolic neural networks. In particular, we will use the hyperbolic feed forward (FF) layer (also known as Möbius layer). Considering the Euclidean case, for a FF layer, we need a matrix $\mathbf{M} : \mathbb{R}^n \to \mathbb{R}^m$ to linearly project the input $\mathbf{x} \in \mathbb{R}^n$ to the feature space $\mathbb{R}^m$, and, additionally, a translation made by a bias addition, i.e., $\mathbf{y} + \mathbf{b}$ with $\mathbf{y}, \mathbf{b} \in \mathbb{R}^m$ and, finally, a pointwise non-linearity $\phi : \mathbb{R}^m \to \mathbb{R}^m$.

Matrix multiplication, bias and pointwise non-linearity are replaced by Möbius operations in the gyrovector space and become:

$$\mathbf{y} = \mathbf{M}^{\otimes_c}(\mathbf{x}) = \frac{1}{\sqrt{c}}\tanh\left(\frac{\|\mathbf{M}\mathbf{x}\|}{\|\mathbf{x}\|}\tanh^{-1}(\sqrt{c}\|\mathbf{x}\|)\right)\frac{\mathbf{M}\mathbf{x}}{\|\mathbf{x}\|} \qquad (3)$$

$$\mathbf{z} = \mathbf{y} \oplus_c \mathbf{b} = \exp_{\mathbf{y}}^c\left(\frac{\lambda_0^c}{\lambda_{\mathbf{y}}^c}\log_0^c(\mathbf{b})\right), \qquad \phi^{\otimes_c}(\mathbf{z}) = \exp_{\mathbf{z}}^c\left(\phi(\log_0^c(\mathbf{z}))\right) \qquad (4)$$

where $\mathbf{M}$ and $\mathbf{b}$ are the same matrix and vector defined above, $c$ is the magnitude of the curvature. Note that when $c \to 0$ we recover the Euclidean feed-forward layer. An interesting property of the Möbius layer is that it is highly nonlinear; indeed the bias addition in hyperbolic space becomes a nonlinear mapping since geodesics are curved paths in non-flat manifolds.

### 3.3 Hyperbolic Compositional Regularization

Armed with the formalism introduced in the previous section, we are ready to formulate our HyCoRe framework, anticipated in Fig. 1. Consider a point cloud $P_N$ as a set of 3D points $\mathbf{p} \in \mathbb{R}^3$ with $N$ elements. We use any state-of-the-art point cloud processing network as a feature extraction backbone $E : \mathbb{R}^{N \times 3} \to \mathbb{R}^m$ to encode $P_N$ in the corresponding feature space. At this point we apply an

exponential map $\exp_{\mathbf{x}}^c : \mathbb{R}^m \rightarrow \mathbb{D}^m$ to map the Euclidean feature vector into the hyperbolic space and then a Möbius layer $H : \mathbb{D}^m \rightarrow \mathbb{D}^f$ to project the hyperbolic vector in an $f$-dimensional Poincarè ball. This is the hyperbolic embedding of the whole point cloud $P_N$, i.e., $\mathbf{z}_{\text{whole}} = H(\exp(E(P_N))) \in \mathbb{D}^f$. We repeat the same procedure for a sub-part of $P_N$, which we call $P_{N'}$ with a number of points $N' < N$, to create the part embedding $\mathbf{z}_{\text{part}} = H(\exp(E(P_{N'}))) \in \mathbb{D}^f$ in the same feature space as before.

We now want to regularize the feature space to induce the previously mentioned properties, namely the part-whole hierarchy and clustering according to the class labels. This is performed by defining the following triplet regularizers:

$$R_{\text{hier}}(\mathbf{z}_{\text{whole}}^+, \mathbf{z}_{\text{part}}^+) = \max(0, -\|\mathbf{z}_{\text{whole}}^+\|_{\mathbb{D}} + \|\mathbf{z}_{\text{part}}^+\|_{\mathbb{D}} + \gamma/N') \tag{5}$$

$$R_{\text{contr}}(\mathbf{z}_{\text{whole}}^+, \mathbf{z}_{\text{part}}^+, \mathbf{z}_{\text{part}}^-) = \max\left(0, d_{\mathbb{D}}(\mathbf{z}_{\text{whole}}^+, \mathbf{z}_{\text{part}}^+) - d_{\mathbb{D}}(\mathbf{z}_{\text{whole}}^+, \mathbf{z}_{\text{part}}^-) + \delta\right) \tag{6}$$

where $\mathbf{z}_{\text{whole}}^+$ and $\mathbf{z}_{\text{part}}^+$ are the hyperbolic representation of the whole and a part from the same point cloud, while $\mathbf{z}_{\text{part}}^-$ is the embedding of a part of a different point cloud from a different class.

The $R_{\text{hier}}$ regularizer in Eq. (5) induces the compositional part-whole hierarchy by promoting part embeddings to lie closer to the center of the Poincarè ball and whole embeddings to be closer to the edge. In particular, we use a variable margin $\gamma/N'$ that depends on the number of points $N'$ of the part $P_{N'}$. This means that shapes composed by few points (hence simple universal shapes) will be far from the whole object representation and with lower hyperbolic norm (near the centre). On the other hand, embeddings of larger parts will be progressively closer to the edge of the Poincarè ball, depend-

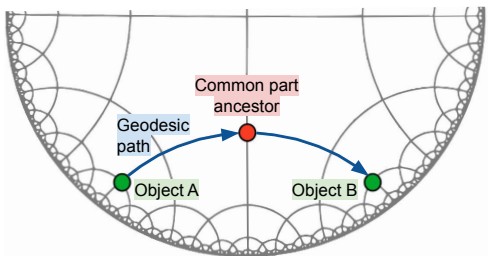

Figure 3: Geodesic path.

ing on the part size. Since geodesics between two points pass closer to the ball center (Fig. 3), this structure we impose to the space allows to visit common part ancestors while traversing a geodesic between two whole objects. This regularization thus mimics a continuous version of a part-whole tree embedded in the Poincarè ball.

The $R_{\text{contr}}$ regularizer in Eq. (6) promotes correct clustering of objects and parts in the hyperbolic space. In particular, parts and whole of the same point cloud are promoted to be close while a part from a different class is mapped far apart with respect to the other whole. It ensures that the parts of a point cloud of a different class are far in terms of geodesic distance. $\delta$ is a margin hyperparameter to control the degree of separation between positive and negative samples.

The two regularizations are included in the final loss in this way:

$$L = L_{\text{CE}} + \alpha R_{\text{contr}} + \beta R_{\text{hier}} \tag{7}$$

where $L_{\text{CE}}$ is the conventional classification loss (e.g., cross-entropy) evaluated on the whole objects. The classification head is a hyperbolic Möbius layer followed by softmax. In principle, one could argue that $L_{\text{CE}}$ could already promote correct clustering according to class labels, rendering $R_{\text{contr}}$ redundant. However, several works [10] have noticed that the Möbius-softmax hyperbolic head is weaker than its Euclidean counterpart. We thus found it more effective to evaluate $L_{\text{CE}}$ on the whole objects only, and use $R_{\text{contr}}$ as a metric penalty that explicitly considers geodesic distances to ensure correct clustering of both parts and whole objects.

At each iteration of training with HyCoRe we sample shapes with a random $N'$ varying within a predefined range. A part is defined as the $N'$ nearest neighbors of a random point. In future work, it would be interesting to explore alternative definitions for parts, e.g., using part labels if available but, at the moment, we only address definition via spatial neighbors to avoid extra labeling requirements.

Table 1: Classification results on ModelNet40. *: re-implemented. **: re-implemented but did not exactly reproduce the reference result.

| Method | AA(%) | OA(%) | Training |
|---|---|---|---|
| *PointNet++[17] | - | 90.5 | supervised |
| *DGCNN[21] | 90.2 | 92.9 | supervised |
| Point Transformer [30] | 90.6 | 93.7 | supervised |
| PA-DGC [31] | - | 93.6 | supervised |
| CurveNet [32] | - | 93.8 | supervised |
| **PointMLP[3] | 91.2 | 93.4 | supervised |
| **PointMLP (voting) | 91.4 | 93.7 | supervised |
| DGCNN+Self-Recon. [33] | - | 92.4 | finetuned |
| DGCNN+STRL [34] | - | 93.1 | finetuned |
| DGCNN+DCGLR [7] | - | 93.2 | finetuned |
| *PointNet++ +PointGLR [5] | - | 90.6 | finetuned |
| **PointNet++ +HyCoRe** | - | 91.1 | regularized |
| **DGCNN +HyCoRe** | 91.0 | 93.7 | regularized |
| **PointMLP +HyCoRe** | **91.7** | **94.3** | regularized |
| **PointMLP +HyCoRe (voting)** | **91.9** | **94.5** | regularized |

Table 2: Classification results on ScanObjectNN.

| Method | AA(%) | OA(%) |
|---|---|---|
| DGCNN[21] | 77.8 | 80.3 |
| SimpleView[4] | - | 80.8 |
| PRANet[35] | 79.1 | 82.1 |
| MVTN[36] | - | 82.8 |
| PointMLP[3] | 84.4 | 86.1 |
| **PointNeXt[37] | 86.4 | 88.0 |
| **DGCNN+HyCoRe** | 80.2 | 82.1 |
| **PointMLP+HyCoRe** | **85.9** | **87.2** |
| **PointNeXt+HyCoRe** | **87.0** | **88.3** |

Table 3: Effectiveness of hyperbolic space.

| | Average Accuracy (%) | | | | |
|---|---|---|---|---|---|
| Dim | 16 | 64 | 256 | 512 | 1024 |
| DGCNN | 76.6 | 77.5 | 77.8 | 76.6 | 76.3 |
| DGCNN+EuCoRe | 78.2 | 78.9 | 79.0 | 78.8 | 79.0 |
| Hype-DGCNN | 76.8 | 75.9 | 76.5 | 76.0 | 77.5 |
| **DGCNN+HyCoRe** | 79.1 | 80.0 | **80.2** | **80.2** | 79.7 |

## 4 Experimental results

### 4.1 Experimental setting

We study the performance of our regularizer HyCoRe on the synthetic dataset ModelNet40 [27] (12,331 objects with 1024 points, 40 classes) and on the real dataset ScanObjectNN [28] (15,000 objects with 1024 points, 15 classes). We apply our method over multiple classification architectures, namely the widely popular DGCNN and PointNet++ baselines, as well as the recent state-of-the-art PointMLP model. We substitute the standard classifier with its hyperbolic version (Möbius+softmax), as shown in Fig. 1. We use $f = 256$ features to be comparable to the official implementations in the Euclidean space, then we test the model over different embedding dimensions in the ablation study. Moreover, we set $\alpha = \beta = 0.01$, $\gamma = 1000$ and $\delta = 4$. For the number of points of each part $N'$, we select a random number between 200 and 600, and for the whole object a random number between 800 and 1024 to ensure better flexibility of the learned to model to part sizes. We train the models using Riemannian SGD optimization. Our implementation is on Pytorch and we use *geoopt* [29] for the hyperbolic operations. Models are trained on an Nvidia A6000 GPU.

### 4.2 Main Results

Table 1 shows the results for ModelNet40 classification. In the first part we report well-known and state-of-the-art supervised models. We retrained PointNet++, DGCNN and the state-of-the-art PointMLP as baselines, noting some documented difficulty [38] with exactly reproducing the

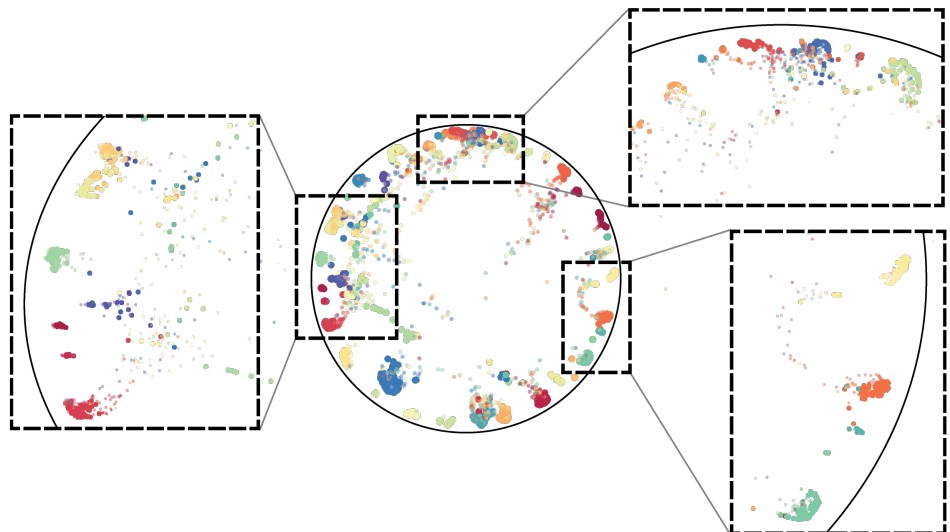

Figure 4: Embeddings produced by the hyperbolic encoder, projected to 2 dimensions with hyperbolic UMAP. Each color represents a class; small points correspond to parts; large points correspond to whole objects. Parts are closer to the center, sitting higher in the hierarchy (whole objects at the border may share a common part ancestor reachable via the geodesic connecting the objects).

Table 4: Classification results when one of the two regularizations is omitted.

|  | AA(%) | OA(%) |
|---|---|---|
| DGCNN | 77.8 | 80.3 |
| DGCNN+$R_{\text{hier}}$ | 77.9 | 80.5 |
| DGCNN+$R_{\text{contr}}$ | 79.2 | 81.6 |
| **DGCNN+HyCoRe** | **80.2** | **82.1** |

Table 5: Performance vs. curvature of the Poincarè Ball

| Average Accuracy (%) | | | | |
|---|---|---|---|---|
| Curvature $c$ | 1 | 0.5 | 0.1 | 0.01 |
| Hype-DGCNN | 76.5 | 76.9 | 76.6 | 76.9 |
| **DGCNN+HyCoRe** | **80.2** | 79.4 | 78.7 | 78.5 |

official results. In addition, the second part of the table reports the performance of methods [33], [34], [7] proposing self-supervised pretraining techniques, after supervised finetuning. Concerning PointGLR [5], the most similar method to HyCoRe, we ensure a fair comparison by using only the L2G embedding loss and not the pretext tasks of normal estimation and reconstruction.

Finally, the last part of the table presents the results with HyCoRe applied to the selected baselines. We can see that the proposed method achieves substantial gains not only compared to the randomly initialized models, but also compared to the finetuned models. When applied to the PointMLP, HyCoRe exceed the state-of-the-art performance on ModelNet40. Moreover, it is interesting to notice that the embedding framework of PointGLR is not particularly effective without the pretext tasks. This is due to the unsuitability of the spherical space to embed hierarchical information, as explained in Sec. 2, and it is indeed not far from results we obtain with our method in Euclidean space.

Table 2 reports the classification results on the ScanObjectNN dataset. Also in this case, HyCoRe significantly improves the baseline DGCNN leading it to be comparable with the state-of-the-art methods such as SimpleView [4], PRANet [35] and MVTN [36]. In addition, PointMLP that holds the state of the art for this dataset, is further improved by our method and reaches an impressive overall accuracy of 87.2 %, substantially outperforming all the previous approaches. Although the authors in [3] claim that classification performance has reached a saturation point, we show that including novel regularizers in the training process can still lead to significant gains. This demonstrates that the proposed method leverages novel ideas, complementary to what is exploited by existing architectures, and it is thus able to boost the performance even of state-of-the-art methods. It is also remarkable that an older, yet still popular, architecture like DGCNN is able to outperform complex and sophisticated models such as the Point Transformer, when regularized by HyCoRe.

Table 6: Hyperbolic Norms of labeled parts from the whole object up to the single parts.

| Table | Plane+uprights | Legs+uprights | Plane | Legs | Uprights |
|-------|----------------|---------------|-------|------|----------|
| 5.32  | 4.56           | 2.08          | 4.07  | 2.05 | 1.99     |

| Aircraft | Wings+tail+engines | Wings+tail | Wings | Fuselage | Tail |
|----------|--------------------|------------|-------|----------|------|
| 4.98     | 4.56               | 4.45       | 4.22  | 3.37     | 2.94 |

In addition, to further prove that enforcing the hierarchy between parts is useful to build better clusters, we show in Fig. 4 a 2D visualization with UMAP of the hyperbolic representations for the ModelNet40 data. Colors denote classes, big points whole objects and small points parts. Besides the clear clustering according to class labels, it is fascinating to notice the emergence of the part-whole hierarchy with part objects closer to the center of the disk. Importantly, some parts bridge multiple classes, such as the ones in the bottom right zoom, i.e., they are found along a geodesic connecting two class clusters, serving as common ancestors. This can happen due to the fact that some simple parts having roughly the same shape appear with multiple class labels during training, and the net effect of $R_{\mathrm{contr}}$ is to position them midway across the classes.

The tree-likeness of the hyperbolic space can be also be seen in the visualization in Fig. 2 (right). There we embed shapes with gradually large number of points up to the whole object made by 1024 points. We can notice that the parts are moved towards the disk edge as more points are added. Furthermore, a quantitative analysis of the part-whole hierarchy is shown in Table 6. Here we calculated the hyperbolic norms of compositions of labeled parts. We can see that, as the parts are assembled with other parts, their hyperbolic norms grow, up to the whole object that is pushed close to the ball edge.

## 4.3 Ablation study

In the following we show an ablation study focusing on the DGCNN backbone and the ScanObjectNN dataset. The dataset selection is motivated by the fact that it is a real dataset, able to provide more stable and representative results compared to ModelNet40.

We first compare HyCoRe with its Euclidean version (EuCoRe) to investigate the effectiveness of the hyperbolic space. The basic principles and losses are the same, but in EuCoRe distances and network layers are defined in the Euclidean space. Table 3 shows the results. With Hype-DGCNN we indicate the hyperbolic version of DGCNN, as represented in Fig. 1, but without any regularization, serving as a baseline to assess the individual effect of the regularizer. We also test the models over a different number of embedding dimensions. We can see that EuCoRe only provides a modest improvement, underlining the importance of the hyperbolic space. We also notice that the hyperbolic baseline struggles to be on par with its Euclidean counterpart, as observed by many recent works [25], [10]. However, when regularized with HyCoRe, we can observe significant gains, even in low dimensions. This also leaves an open research question, about whether better hyperbolic baselines could be built so that HyCoRe starts from a less disadvantaged point.

In Table 4 we ablate HyCoRe by removing one of the two regularizers. We can see that the combination of the two provides the overall best gain.

In order to study the effect of different space curvatures $c$, Table 5 evaluates HyCoRe from the standard curvature 1 down to 0.01. We remark that some works [25],[39], report significant improvements when $c$ is very low (e.g., 0.001), but this is counter-intuitive since the hyperbolic space then resembles an almost flat manifold. On the contrary, we do see improved results at higher curvatures.

Since HyCoRe constrains the network to learn the relations between parts and whole object, we claim that, at the end of the training process, the model should be better able to classify coarser objects. In Figs. 5a and 5b we show the test accuracy of DGCNN on ModelNet40, when presented with a uniformly subsampled point cloud and with a small randomly chosen and spatially-contiguous part, respectively. Indeed, we can notice that HyCoRe provides a gain up to 20 percentage points for very sparse point clouds, and is also able to successfully detect the object from smaller parts. For a fair comparison, we also report the baseline DGCNN with training augmented by random crops

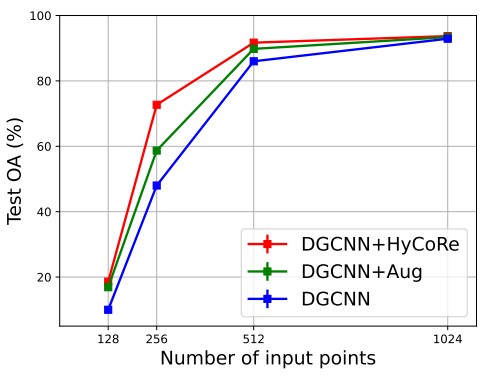 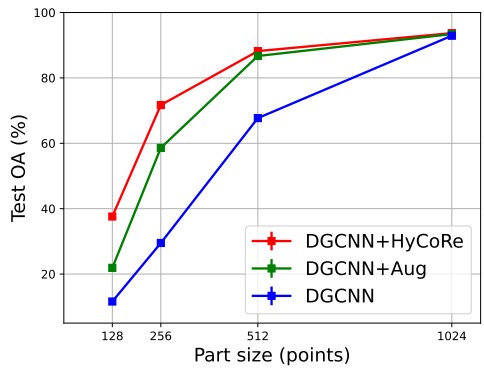

(a) Subsampled input. HyCoRe is more robust when the point cloud has coarser sampling.

(b) Parts with different size. HyCoRe better detects objects from only a small part.

Figure 5: Test inference of DGCNN on ModelNet40.

of parts. Even though the augmentation is useful to improve accuracy, HyCoRe is more effective demonstrating the importance of compositional reasoning.

## 5  Conclusions

Although deep learning in the hyperbolic space is in its infancy, in this paper we showed how it can successfully capture the hierarchical nature of 3D point clouds, boosting the performance of state-of-the-art models for classification. Reasoning about the relations between objects and the parts that compose them leads not only to better results but also more robust and explainable models. In the future, it would be interesting to explore different ways of defining parts, not based on spatial nearest neighbors but rather on more semantic constructions. One important extension is to adapt HyCoRe to segmentation. Since segmentation aims to classify single points and the corresponding parts, contrary to classification, the parts embeddings should be placed on the boundary of the Poincarè Ball, where there is more space to correctly cluster them, and the whole objects (made by composition of parts) near the origin. We could exploit the label of the parts to this end or investigate unsupervised settings where the part hierarchy emerges naturally.

## Acknowledgments and Disclosure of Funding

Computational resources were provided by HPC@POLITO, a project of Academic Computing within the Department of Control and Computer Engineering at the Politecnico di Torino (http://www.hpc.polito.it). This research received no external funding.

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
