# Rethinking the compositionality of point clouds through regularization in the hyperbolic space - Supplementary material

**Antonio Montanaro**[*]
Politecnico di Torino, Italy
antonio.montanaro@polito.it

**Diego Valsesia**
Politecnico di Torino, Italy
diego.valsesia@polito.it

**Enrico Magli**
Politecnico di Torino, Italy
enrico.magli@polito.it

## 1 Geodesic path

We show how the geodesic path between two objects traverses common part ancestors. In particular, we start from the hyperbolic embedding of object A $\mathbf{z}_A = H(\exp(E(P_N^A)))$, and trace the geodesic to the hyperbolic embedding of object B $\mathbf{z}_B = H(\exp(E(P_N^B)))$. For a number of points on the geodesic we look for the nearest neighbors (hyperbolic distance) among the embeddings of objects and parts in the dataset. To this aim, we use the parametric version of the geodesic, defined as:

$$\gamma_{\mathbf{z}_A \to \mathbf{z}_B}(t) = \mathbf{z}_A \oplus_c (-\mathbf{z}_A \oplus_c \mathbf{z}_B) \otimes_c t, \gamma_{x \to y} : \mathbb{R} \to \mathbb{D}_c^n, \tag{1}$$

where $t$ is the step size along the geodesic, such that $\gamma(t=0) = \mathbf{z}_A$ and $\gamma(t=1) = \mathbf{z}_B$. The Mobius operations, i.e. the addition and the scalar multiplication, are defined in the gyrovector space through the following formulas:

$$\mathbf{x} \oplus_c \mathbf{y} = \frac{(1 + 2c\langle \mathbf{x}, \mathbf{y} \rangle + c\|\mathbf{y}\|^2)\mathbf{x} + (1 - c\|\mathbf{x}\|^2)\mathbf{y}}{1 + 2c\langle \mathbf{x}, \mathbf{y} \rangle + c^2\|\mathbf{x}\|^2\|\mathbf{y}\|^2}, \tag{2}$$

$$t \otimes_c \mathbf{x} = (1/\sqrt{c}) \tanh(t \tanh^{-1}(\sqrt{c}\|\mathbf{x}\|)) \frac{\mathbf{x}}{\|\mathbf{x}\|}, \tag{3}$$

We analyze different paths in the hyperbolic 256-dimensional space for DGCNN regularized by our method HyCoRe. A sketch of the geodesic interpolation is represented in Fig. 1.

In Fig. 2 we show three geodesics from different pairs of objects. We can see that, near the whole objects, the parts are bigger and specific to that class, while in the midpoints of the geodesic, common part ancestors emerge and are shared by the two objects.

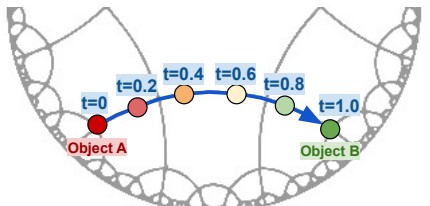

Figure 1: Geodesic path.

Furthermore, since geodesics length changes according to the connecting objects, we add a geodesic in Fig. 3 for two objects of the same class. Even in this case common parts are visible. This additional analysis reinforces our claim that the tree-like structure of point cloud data is preserved at different hierarchies.

---

[*]Code of the project: https://github.com/diegovalsesia/HyCoRe

36th Conference on Neural Information Processing Systems (NeurIPS 2022).

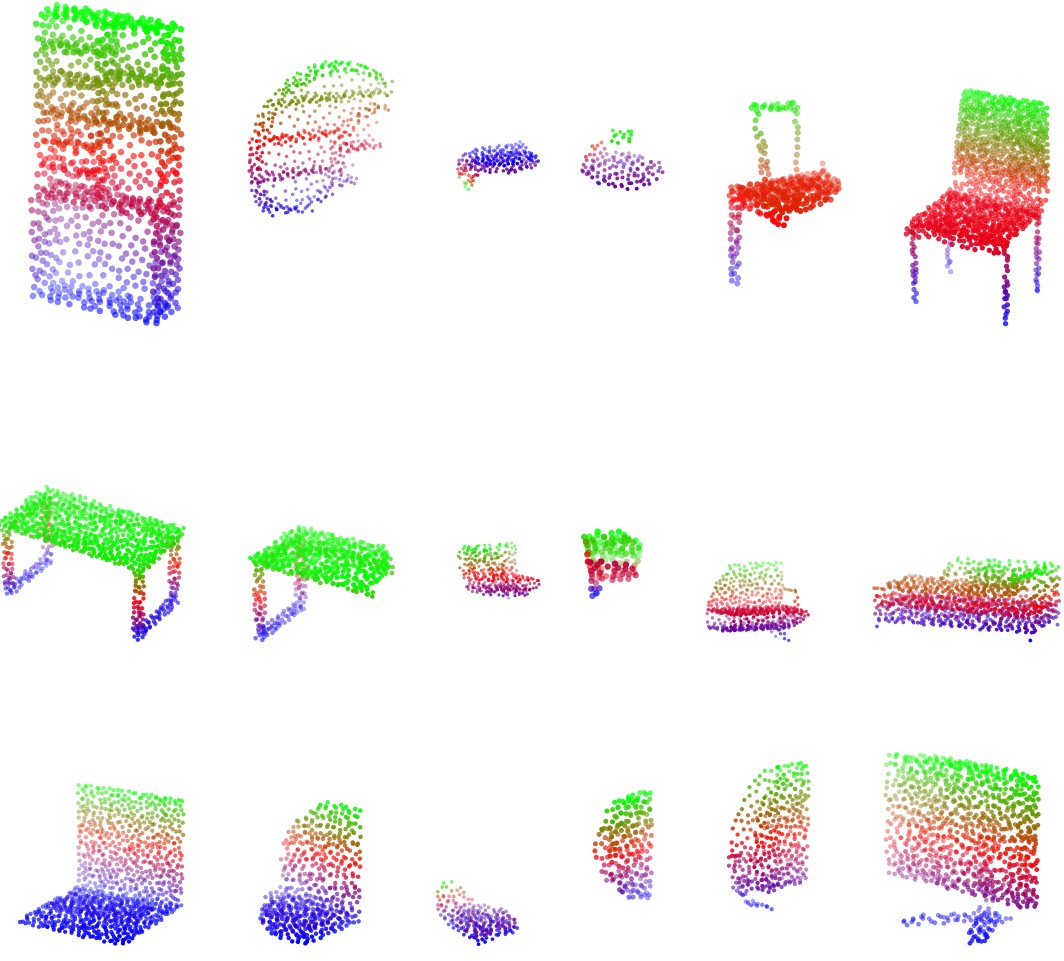

Figure 2: Hyperbolic nearest neighbors of points along a geodesic from the embedding of object A and object B (ModelNet40) using our DGCNN+HyCoRe. As we approach to the midpoint of the geodesic, smaller parts are encountered, indicating common ancestors shared by the two objects.

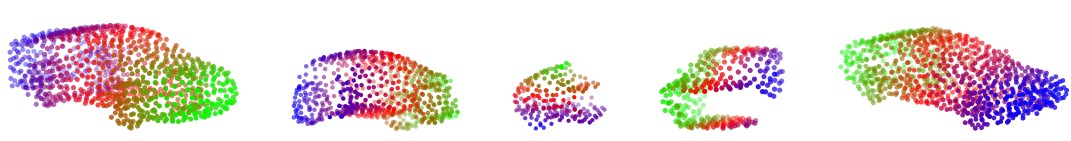

Figure 3: Interpolating a geodesic across two objects belonging to the same class leads to consistent parts that become smaller and more general, respecting the tree-like structure induced by our HyCoRe.

## 2 Embedding visualization

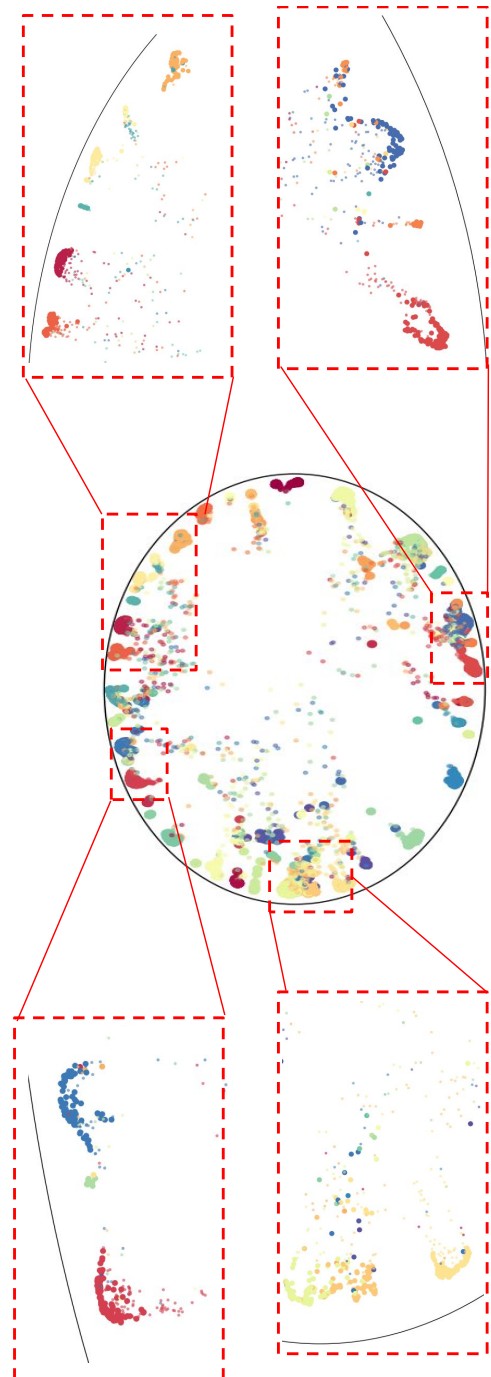

Figure 4: Embeddings produced by the hyperbolic encoder, projected to 2 dimensions with hyperbolic UMAP. Color=class; small points = parts; large points = whole objects.