# OpenReview forum: "Rethinking the compositionality of point clouds through regularization in the hyperbolic space"
_NeurIPS.cc/2022/Conference — NeurIPS 2022 Accept_

### Official Review · Reviewer_p5Dv · 2022-07-11

**Rating:** 5
**Confidence:** 4
**Soundness:** 4 excellent
**Presentation:** 3 good
**Contribution:** 2 fair

**Summary:**

The authors present a novel representation for latent space of point clouds to capture and take advantage of the hierarchical nature of the underlying shapes. The proposed method encourages point clouds to be projected to a hyperbolic space, where common simple structures are forced to the center of the space, and the complex shapes are moved to the edge.

To achieve this they propose a module that can be added onto existing frameworks. The module takes the feature vector of a set of points representing a shape and projects it to a Poincarè ball. To achieve the proposed hierarchical representation, they also extract partial point cloud from the same shape and other shape as input and use it in the regularization term. The constraints push the partial point clouds to the center, while keeping the parts of the same point cloud close and pushing back parts from other point clouds.

The regularization strategy establishes a more effective latent feature space, as seen from the classification accuracies on the benchmark datasets ModelNet40 and ScanObjectNN, as well as from the visualization of the resulting space, which shows features from different classes are kept apart.

**Questions:**

I wondered if the partial point clouds would contaminate the feature space of the backbone network. Have the authors tried freezing the backbone networks so that the feature calculation backbone network is not affected by the triplet loss?

I also wondered what would happen if the partial point clouds created in each epoch for the constraints were centralized. It would seem to lead to a more robust common part ancestor, but leading to difficulty in establishing hierarchy between whole and partial shapes.

If segmentation was conducted in the suggested way in the section above, how would the results be? It is a shame that partial information is only utilized to map the point clouds in the hyperbolic space. All the efforts to include such data do not seem to be fully utilized.

Have the authors attempted to use other sets of data for visualization? Although slightly out of context, it may have been better to use a more simple set of targets for the visualization task, such as humans in different poses. Human models have parts that can be used as the partial data, and commonality and transformation are easier to observe.

**Limitations:**

The authors do mention the procedure of preparing the partial shapes to be the main limitation of the proposal. The method currently takes a random point and collects $N'$ nearest points to define one partial shape. Using segment data could improve the partial representation of the proposed feature space.

Also, I believe lack of a framework to use the obtained feature for shape segmentation is also a drawback, as the authors go so far as to including parts of shapes as input to the proposed framework.

**Strengths And Weaknesses:**

### Strengths

The representation is very novel and interesting, as it attempts to take advantage of the hyperbolic space in order to establish hierarchical relationships that starts from common simple parts to wholistic characteristic shapes. The authors thoroughly explain the geometric background of the proposal, and define a sound and practical constraints that drives data to be in the desired respective positions in the hierarchy.

The training procedure of preparing partial shapes to establish this hierarchy is also unique. The resulting feature space, as illustrated in Fig. 4, suggests that the constraints on the feature space have effectively mapped the latent features of even partial data to the desired locations.

The fact that the proposed module can be placed on top of existing point cloud analysis backbone networks is a huge advantage. By employing this representation, the conventional methods gain massive boosts in classification accuracy, which can be observed from Tables 1 and 2.

### Weaknesses

The training procedure seems to require effort. From the text, the partial shapes have to be prepared at each epoch just to calculate the triplet loss defined in the paper. As the process is rather random, we can easily imagine the training process taking much longer than most existing frameworks. The burden of the training process seems to be excluded in the experimental section.

It is doubtful whether the claimed hierarchy is actually achieved by the proposal. The colormap in Fig. 4 seems to be well-organized near the edge of the ball, but seems rather random near the middle. The interpolation results in Fig.2 of the appendix, despite the efforts to incorporate various partial shapes in the training phase, also isn't as convincing as desired. They are definitely smaller in size, but the claimed shape commonality is difficult to observe.

Despite the efforts to consider parts of shapes, it is disappointing that the authors do not include any point segmentation task, which is deeply related to local information of shapes, in the evaluation. One can easily imagine using the $\mathbf{z}_{whole}$ feature as the global feature, which can be concatenated with conventional point-wise feature to conduct such task.

---

> ### Author Response · Authors · 2022-07-29
> **Response to reviewer p5Dv**
>
> We thank the reviewer for their positive assessment of our work. In the following , we address the raised concerns on a point by point basis.
>
> > 1. “The training procedure seems to require effort...”
>
> We thank the reviewer for this comment. We conducted a test to evaluate the computational time. Although this does not change in inference, the regularization in the training causes a delay. We estimate an incremental training time of the 60% over the training without regularization. We would also like to note that this time is still small when compared to self-supervised pretraining techniques, like contrastive learning, that require training for thousands of epochs and huge batch sizes.
>
> > 2.”It is doubtful whether the claimed hierarchy is actually achieved by the proposal...”
>
> As we discussed in the response to the other reviewers, we can experimentally observe that the geometric placement of objects and parts approximately follows our description. This means that geodesics approximately embed tree distances. Moreover, while we do not require nor exploit semantic parts, the toy experiment performed for reviewer r1Np interestingly shows that even embeddings of semantic parts and compositions thereof follow the described hierarchy.
>
> > 3. “Despite the efforts to consider parts of shapes, it is disappointing that the authors do not include any point segmentation task.One can easily imagine using the feature as the global feature, which can be concatenated with conventional point-wise feature to conduct such task.”
>
> As we discuss in the responses to other reviewers, we deliberately omit a discussion on part segmentation to focus on whole-object classification. The reason is that there are significant differences in the segmentation task, which lead to significant differences in methodology and presenting both tasks would not fit the limits of the paper. In particular, for the part segmentation task one would like the embeddings of the parts to lie close to the Ball boundary, so that they can cluster more effectively as they are used to directly derive the labels. This is in contrast with the object classification problem where we want the embeddings of the whole objects to be on the boundary. This means that, for part segmentation, the optimal hierarchy to be induced is the opposite of what is desired for classification. Fitting a thorough discussion on this matter with supporting analysis would be impossible within the constraints of the paper, so we chose to limit the scope to classification and present this in future work. We also do not believe that concatenating the global embedding z_whole to the point feature would carry relevant information for the part segmentation task, especially because part semantics are never used nor observed in the regularizer.
>
> Questions:
>
> > 1. “I wondered if the partial point clouds would contaminate the feature space of the backbone network. Have the authors tried freezing the backbone networks so that the feature calculation backbone network is not affected by the triplet loss?”
>
> In the Table 3 we train Hype-DGCNN, a DGCNN with hyperbolic projection without the regularizations. The model has a performance loss, and it improves in accuracy when independently adding one of the two regularizers and finally gets the best accuracy when both regularizers are used (Table 4). We do not think that freezing the backbone would be beneficial as this would prevent it from learning a feature extractor that follows the desired hierarchy.
>
> > 2.”I also wondered what would happen if the partial point clouds created in each epoch for the constraints were centralized.”
>
> We thank the Reviewer for this interesting comment. We did not centralize the parts, but since this seems a good idea, we will perform an experiment to see if any improvements can be obtained.
>
> > 3. “If segmentation was conducted in the suggested way in the section above, how would the results be? It is a shame that partial information is only utilized to map the point clouds in the hyperbolic space. All the efforts to include such data do not seem to be fully utilized.”
>
> We want to remark that in our work we did not use labeled parts to avoid extra labeling requirements and work within the standard framework for object classification. For more detail please see the previous response regarding why we have not addressed segmentation..
>
> > 4. “Have the authors attempted to use other sets of data for visualization? ”
>
> We thank the reviewer for this constructive suggestion. We did not include this kind of data for visualization, but we could add some examples of this type in the final version.

---

> > ### Author Response · Authors · 2022-07-29
> > **Response to reviewer p5Dv - part 2**
> >
> > >Limitations:
> > "The authors do mention the procedure of preparing the partial shapes to be the main limitation of the proposal. The method currently takes a random point and collects  N′ nearest points to define one partial shape. Using segment data could improve the partial representation of the proposed feature space. Also, I believe lack of a framework to use the obtained feature for shape segmentation is also a drawback, as the authors go so far as to including parts of shapes as input to the proposed framework."
> >
> > We also believe that segmentation is an important framework to be considered, and indeed we are currently working on this. As discussed in the replies to the other reviewers, we have found that the optimal regularization differs between classification  and segmentation. However this study is in progress and will be reported in a separate paper.
> > Regarding definitions of parts alternative to N’ spatial nearest neighbors, we agree that it could be interesting to test the use of segmentation labels. However, we opted not to do that to avoid extra labeling requirements and fit the standard classification setting. We are also exploring the possibility of defining parts as nearest neighbors in feature space in order to exploit self-similarities.

---

> > > ### Comment · Reviewer_p5Dv · 2022-08-09
> > > **Regarding the rebuttal**
> > >
> > > Thank you for clarifying the concerns!
> > > The questions were mostly clarified by the rebuttal.
> > > I do however still have concerns regarding my points 1 and 3.
> > >
> > > Regarding point 1, about the method requiring extra effort:
> > > after further reading, I agree with Reviewer r1Np that the comparison may not be completely fair.
> > > From my perspective, it seems that the proposed strategy is a way of data augmentation, therefore comparison with backbone models with the partial data also as training data should clarify the true advantages of the proposal.
> > >
> > > Regarding point 3 about segmentation:
> > > I find the reasons provided by the authors not too convincing. As the point cloud analysis is moving onwards to solving more challenging scenarios such as achieving rotation-invariance and noise-resistance, I believe the proposal require some versatility if it were to be proposed not to address these issues. It may not be in the interest of the authors, but can't the global feature be extracted by simply replacing the classification head with the pointwise segmentation head? Even thought there might not be much contribution technically, such results seem necessary to be presented as a point cloud analysis paper.

---

> > > > ### Author Response · Authors · 2022-08-09
> > > > **Further clarifications**
> > > >
> > > > Regarding comparisons against using partial data as training samples, our preliminary experiments did not show any significant advantage in using this strategy. Exploiting them in the regularized way of EuCoRe/HyCoRe appeared to be the best course of action. However, we also remark that our comparisons are not purely against backbone models. As shown in Table 1, we compare against self-reconstruction [33], STRL [34], DCGLR [7], PointGLR [5] which are all self-supervised pretraining techniques that can be interpreted as generating more or less sophisticated "augmentations". These can be considered as the suggested data augmentation baseline.
> > > >
> > > > Regarding segmentation, as mentioned to reviewer 45jd, it could be possible to include some preliminary results replacing the classification head with the segmentation head, and regularization with the inverted hierarchy, but we feel this extends the scope of the paper too much and does not allow for a proper discussion of the differences between classification and segmentation in the page limits of a conference paper.  The reviewer also, interestingly, mentions the need for robust techniques. Indeed, this was also a concern of ours and the experiments in Figs. 5 and 6 show how the proposed method provides substantial gains in presence of subsampled or partial input point clouds.

---

> > > > > ### Comment · Reviewer_p5Dv · 2022-08-09
> > > > > **Thank you for further clarification**
> > > > >
> > > > > I appreciate the further feedback!
> > > > >
> > > > > Again, I have to reiterate that both points go back to testing against backbone networks reinforced by partial data. Fig 5,6 I presume are comparison against non-augmented backbones, so although impressive, has little evidence supporting that the proposal is not a mere data augmentation strategy.
> > > > >
> > > > > I will take these into account during the final disucussions.
> > > > >
> > > > > Thank you for the feedback!

---

> > > > > > ### Author Response · Authors · 2022-08-09
> > > > > > **Latest results on augmentation**
> > > > > >
> > > > > > We report the results of using partial shape augmentation to reinforce the DGCNN backbone. The training generated partial shapes with a random number of points to supplement the full shapes. As mentioned above, the improvement over the non-augmented baseline is not significant. However, we do observe improvements in the robustness tests, albeit not as large as those obtained by the proposed method. We will update the robustness results in the final paper.
> > > > > >
> > > > > > ModelNet40 (OA%)
> > > > > >
> > > > > > DGCNN: 92.9
> > > > > > DGCNN+aug.: 93.1
> > > > > > DGCNN+HyCoRe: 93.7
> > > > > >
> > > > > >
> > > > > > Robustness tests (new versions of Figs.5,6):
> > > > > >
> > > > > > https://ibb.co/kxL97gw
> > > > > >
> > > > > > https://ibb.co/Z8ZRQBm

---

### Official Review · Reviewer_v4Kn · 2022-07-14

**Rating:** 5
**Confidence:** 4
**Soundness:** 2 fair
**Presentation:** 2 fair
**Contribution:** 2 fair

**Summary:**

This paper proposes to embed the features of a point cloud classifier into the hyperbolic space and explicitly regularize the space to account for the part-whole hierarchy. To do this, it employs a hyperbolic neural network [10] and introduces losses to regularize the part-whole hierarchy.


**Questions:**

(1) further discussion of the definition of the part-whole hierarchy and its difference to object-part hierarchy usually defined in literature.

(2) comments on if $R_\text{hier}$ is sufficient to regularize hierchy with multiple levels, and also supply the result of DGCNN+HyCoRe without  $R_\text{hier}$

(3) improve visualization of figure 4, give more visual example of the pointcloud & embedding pairs.

See discussions above for explanations.

**Ethics Review Area:**

["I don’t know"]

**Limitations:**

I could not find a limitation section in the draft.

**Strengths And Weaknesses:**

Strengths:
It seems the proposed use of hyperbolic neural network with regularization is able to improve accuracy of different point cloud classification backbones. The experiments demonstrated the benefit of the proposed work.


Weakness:

The intuition of the proposed method looks handwavy to me.

In literature, when talking about part-whole hierarchy, it almost always refers to a single object consists of multiple subparts, and the subparts further dissolve into smaller parts. The parts need not to be the same and most times they are assumed to be a mixture of different parts. However, in this work, it is in an unusual opposite direction, i.e. different objects share a single “atom” part and the full objects are “grown” piece by piece in sequential order (as shown in figure 2). Such definition of the hierarchy looks suspicious to me, as there shouldn’t be an unique sequential order / hierarchy to define how an object instance is composed in piecewise order, and it makes less sense to require different objects correspond to a single “common part ancestor”. This is in contrast to embedding WordNet (in the original hyperbolic neural network paper) where the word hierarchy is uniquely defined according to categorical relationships.

Also, the way the author designed the contrastive and hierarchical loss (equation 5) is not fully justified by the author’s definition of part-whole hierarchy. The loss in equation 5 only enforces the embedding of the whole object to differ from the embedding of parts, in other words it only distinguishes the last level of hierarchy against the rest levels, but there is no regularization to distinguish subparts between intermediate levels.  Looking at the ablation in Table 4, it does seem that the hierarchical loss make little difference to baseline. Could the author also supply a version of DGCNN+HyCoRe without $R_\text{hier}$? The visualization in Figure 4 is not that helpful either, it is hard to tell the size of each points when everything is densely packed, and not straightforward to see the quality of the learned “part-whold hierarchy” other than the clustering.

Overall, I feel “hierarchy” may not be a good explanation to what the authors actually did. Probably just part-whole contrastive learning (without the hierarchical part) is more appropriate.

---

> ### Author Response · Authors · 2022-07-29
> **Response to reviewer v4Kn**
>
> We thank the reviewer for their positive assessment of our work. In the following , we address the raised concerns on a point by point basis.
>
> > 1. “Such definition of the hierarchy looks suspicious to me, as there shouldn’t be an unique sequential order / hierarchy to define how an object instance is composed in piecewise order, and it makes less sense to require different objects correspond to a single “common part ancestor””
>
> We agree with the reviewer that there could exist a multiplicity on how an object instance is composed in piecewise order, but the same multiplicity could exist even in the opposite hierarchy, e.g. an object ancestor could be split in leaves by parts that belong to other objects. As discussed in the response to other reviewers, this apparently unintuitive hierarchy is suited to exploiting the properties of the hyperbolic space for the whole point cloud classification problem we are tackling.
> The idea of using this kind of hierarchy is that we want the final leaves of the tree, i.e. the whole objects to nicely cluster according to the different classes. To do this, we embed small parts with generic shape near the centre (where the poincare ball has “limited volume”)  and more and more specific parts or whole objects near the ball boundary (where the volume has increased exponentially as function of radius and allows comfortable clustering). The method that ensures this propagation is the variable margin in Equation (5). We remark that we do not require nor exploit a semantic definition for the parts, as we do not attempt to construct a model for part segmentation or a constructive generative process. Nevertheless, the toy experiment we performed for reviewer r1Np shows that even embeddings of semantic parts and compositions thereof follow the claimed hierarchy. Finally, we remark that we do not address part segmentation, as it would require a lengthy discussion and more analysis that cannot fit the length of the paper, but that task would require the opposite hierarchy to be optimally regularized (also see response to reviewer 45jd).
>
> > 2. “Also, the way the author designed the contrastive and hierarchical loss (equation 5) is not fully justified by the author’s definition of part-whole hierarchy. ”
>
> Equation (5) can promote different levels of embeddings from the center to the edge thanks to variable margin γ/N’ where N’ is the number of points in the part. Being dependent on N’, when a small part is embedded in the feature space, the margin is stronger and enforces the difference between the two norms to be large, landing the part near to the centre while the whole object is always pushed on the edge by the other regularizer (Equation (6)).On the contrary when the part is bigger the margin is weaker and the part can be far from the centre but above the object. The variability of N’ guarantees a continuous hierarchy from the center to the edge according to the part size. Notice that we use part size N’ as a proxy for part specificity that does not require semantic labels.
>
> > 3. “Overall, I feel “hierarchy” may not be a good explanation to what the authors actually did. Probably just part-whole contrastive learning (without the hierarchical part) is more appropriate.”
>
> We use the term hierarchy since the hyperbolic triplet loss was used in other works to define the tree structure for various data (images, words…) with the same claim to embed this hierarchy in a continuous space. We agree that there are similarities with a part-whole contrastive learning but a main concept differentiates our work from it. The variable margin guarantees different parts to be located at different levels in the space, while the contrastive loss between parts and whole objects ensures the objects to be located on the edge and be connected by ancestor. We can also see in the ablations in Table 4 that the hierarchy regularizer synergizes well with the contrastive learning. A pure contrastive approach would not perform as well (third row of Table 4).
>
>
> Questions:
>
> > 1. “further discussion of the definition of the part-whole hierarchy and its difference to object-part hierarchy usually defined in literature.”
>
> Answered and commented in point 1 above
>
> > 2. “comments on if Rhier is sufficient to regularize hierchy with multiple levels, and also supply the result of DGCNN+HyCoRe without Rhier”
>
> Answered and commented in point 2 above. The requested DGCNN+HyCoRe without Rhier is reported in the third row of Table 4. It is true that Rhier by itself can only marginally improve the baseline but it synergizes well when combined with Rcontr so that 1.0 AA points are gained by the synergy with respect to DGCNN+HyCoRe without Rhier.
>
> > 3. “improve visualization of figure 4, give more visual example of the pointcloud & embedding pairs.”
>
> We thank the reviewer for this suggestion. We will do our best to improve the visualization of figure 4 and add more examples of objects pairs in the final version of the paper.

---

### Official Review · Reviewer_GCkG · 2022-07-16

**Rating:** 7
**Confidence:** 3
**Soundness:** 3 good
**Presentation:** 3 good
**Contribution:** 4 excellent

**Summary:**

This paper presents a method for promoting the part-whole hierarchy in the learned feature space. In particular, it proposes to embed the features of a point cloud encoder into hyperbolic space. The part-whole hierarchy is enhanced through an explicit regularizer. Such a key idea is backed by the theory that the hyperbolic space (space with negative curvature) is the only space that can embed tree structures with low distortion. A regularizer layer is proposed for supervised training of point cloud classification models. It can be applied to existing architectures with a simple modification. Performance boost across a number of popular architectures is reported in the experiments.

**Questions:**

* It is not clear to me why Equation (5) can promote part embeddings to lie closer to the center while whole embeddings to the edge. The main reason is that Equation (5) only specifies constraints on the distance between the embedding of whole and part. It only encourages the part and the whole feature to stay apart. Why the part feature (instead of the whole feature) is guaranteed to be pushed to the center? What is the technical insight behind it?

* Why hierarchy learning can be beneficial to the task of classification? In the end, only the whole shape is used for classification, and the parts are not used as the input. What is the advantage of the proposed method over a naive method that simply applies contrastive learning on the feature space of the whole shape?

**Limitations:**

No limitations or potential negative societal impact are provided in the paper. I would recommend the authors provide some failure cases or limitations in the rebuttal (if any).

**Strengths And Weaknesses:**

### Strengths

*  The idea of promoting a part-whole hierarchy in the hyperbolic space provides a novel perspective for learning discriminative features of the point cloud, which I believe is beneficial to the community. Though it is only verified in the task of classification, I think it could be valuable for more tasks that require in-depth analysis of part hierarchies, e.g. matching incomplete point cloud, point cloud generation from parts, etc.

* The experiments have shown steady performance improvement by applying the proposed method to existing mainstream backbones.

* Nice visualization of the formed feature space for a better understanding of the effectiveness of the proposed approach.

* Code is provided for reproduction.

### Weaknesses

* Since hyperbolic learning is new to the point cloud analysis community, more insights on the technical part should be given to assist the audience to comprehend why the proposed method could work as expected. I would point out what should be improved in the questions section.

* The paper claims the proposed method can be applied to any existing architecture. However, the result section only shows it is applied to a small subset of methods that are compared. This makes me wonder if the results are cherry-picked. Can it provide a performance boost to arbitrary architecture?

---

> ### Author Response · Authors · 2022-07-29
> **Response to reviewer GCkG**
>
> We thank the reviewer for their positive assessment of our work. In the following , we address the raised concerns on a point by point basis.
>
> > 1.”Since hyperbolic learning is new to the point cloud analysis community, more insights on the technical part...”
>
> We thank the reviewer for this constructive comment. We will add more insights and explanations on the hyperbolic learning in the final version of the paper, highlighting how the new projections on non Euclidean spaces can be beneficial for point cloud data.
>
> > 2. “...However, the result section only shows it is applied to a small subset of methods that are compared...”
>
> The architectures to which we applied the proposed method have been chosen with a clear rationale. First, PointNet++ has been one of the first architectures to process point clouds and it is still widely used due to its simplicity and relatively low computational complexity. Then, DGCNN is most important architecture belonging to the class of graph neural networks and it is widely used as a benchmarking baseline. Finally, PointMLP was the state-of-the-art for point cloud classification at the time of writing and serves the purpose of demonstrating that the proposed technique can further enhance even the best performing model.
> During the revision period, a new architecture (PointNext) has been posted as a preprint, further improving the state of the art. We tested our method on PointNext, still confirming relevant gains as reported below, and confirming the generality of our approach. The following results have been obtained on ScanObjectNN PointNext OA 88.0%, PointNext+HyCoRe OA 88.3%. PointNext AA 86.4%, PointNext+HyCoRe AA 87.0%. This result would place PointNext+HyCoRe as the state-of-the-art for the challenging ScanObjectNN dataset.
>
> Questions
> > 1. “It is not clear to me why Equation (5) can promote part embeddings to lie closer to the center while whole embeddings to the edge...”
>
> Equation (5) can promote different levels of embeddings from the center to the boundary thanks to variable margin γ/N’ where N’ is the number of points in the part. Being dependent on N’, when a small part is embedded in the feature space, the margin is stronger and enforces the difference between the two norms to be large, landing the part near the center while the whole object is always pushed on the boundary by the other regularizer (Equation (6)). On the contrary, when the part is bigger the margin is weaker and the part can be far from the center but above the object. The variability of N’ guarantees a continuous hierarchy from the center to the boundary according to the part size.
>
> > 2. “Why hierarchy learning can be beneficial to the task of classification? In the end, only the whole shape is used for classification, and the parts are not used as the input. “
>
> The idea is that building a tree structure from generic shapes to specific parts/objects  induces  the leaves/whole-objects  to be more separate according to the parts they are composed of, eventually leading to better clustering of the objects. The benefits of HyCoRe are also visible in figures 5 and 6 where the network regularized by HyCoRe is more robust  to input subsampling or partialization.
>
> > ”What is the advantage of the proposed method over a naive method that simply applies contrastive learning on the feature space of the whole shape?”
>
> The advantage of the proposed method is strongly related to exploiting the tree likeness of hyperbolic space. Simple contrastive learning in hyperbolic space can lead to vectors spreading out and accumulating on boundary, also causing numerical problems. Instead, the regularization in Equation (5) prevents this phenomenon and helps to connect detailed parts with the corresponding objects and general parts at the root of different objects. We think and show some visual examples that the induced regularization generates better clusters among classes. This is also confirmed by the results in Table 4 which highlights the synergies between Rhier and Rcontr.
>
> >Limitations
>
> We do not see a clear way for the work to have negative societal impact. Regarding limitations, as discussed with the other reviewers, the fact that the method cannot simultaneously be optimal for classification and part segmentation can be a limiting factor, as it requires a different regularizer. However, the study on part segmentation is still in its infancy and due to the complexity of the matter, part segmentation is outside the scope of this work.

---

> > ### Comment · Reviewer_GCkG · 2022-08-09
> > **Post rebuttal**
> >
> > Thanks for the author's response! It has addressed my concerns. I will keep my rating as Accept.

---

### Official Review · Reviewer_r1Np · 2022-07-18

**Rating:** 4
**Confidence:** 3
**Soundness:** 2 fair
**Presentation:** 3 good
**Contribution:** 2 fair

**Summary:**

The paper proposed to utilize hyperbolic space to learn part-whole hierarchy for 3D point clouds. The main idea is regularizing in the hyperbolic space if part or whole are representing same object, and if they are from different objects, the regularization is pushing them to larger distance (similar to triplet loss, or contrastive loss). The paper applied the proposed regularization to many different network architecture (e.g. DGCNN, PointNet++, PointMLP, etc) and improved the baseline across two datasets (ModelNet40 and ScanObjectNN).

**Questions:**

See Weakness section above.

**Ethics Review Area:**

["I don’t know"]

**Limitations:**

See weakness section above.

**Strengths And Weaknesses:**

Strength:

1. Utilizing hyperbolic space for regularizing part-whole hierarchy is a new idea. The geodesic distance in the hyperbolic space naturally suitable for the tree structure of part-whole hierarchy and defining the regularization on the part-whole space makes more sense comparing with defining on the Euclidean space. (although I have some questions below)

2. The proposed regularization is agnostic to different network architectures, and the  paper experimented on multiple  different backbone for point cloud classification, each achieved improvement over compared baselines on two dataset.

3. The paper is well written and easy to understand.

Weakness:

1. (minor point) The definition of part-whole hierarchy seems controversial. In the paper, the part is defined as the ancestor of the whole object, and different object can share same ancestor. This is not intuitive, as one object is composed of multiple **different** parts, that means, if looking from the path from part to whole, it's not a tree structure  (one child node is strictly below one parent node). An intuitive way of defining part-whole hierarchy is the whole is the ancestor of the parts and parts can be share with many different objects. This is also how PartNet [a] (A dataset for part-whole hierarchy for 3D object) is created, and also the part-whole hierarchy mentioned in the seminal work [b].

2. Overclaiming. I don't agree with the author the paper is learning to "promotes the part-whole hierarchy of compositionality in the hyperbolic space" Specifically, the only thing the paper is proposing a regularization on the feature space, therefore how the object is composed with different part is not clear, there's no explicit way of representing object as a hierarchical tree using the proposed method, not mentioning the compositionality. Besides, the way the paper is defining "parts" is through subsampling a small local region of the object (Line 188), this is not a semantic part and is only a subgroup of the object, and also the definition of part-whole hierarchy is controversial as above

3. The comparison is not a fair comparison. I appreciate the paper compared with multiple network architectures and see the improvement over compared baselines. However, since the main idea is proposing a regularization loss in the hyperbolic space, a fair comparison should be apply the same regularization loss, but in the Euclidean space. The compared results is only using supervised loss for pretraining and fine-tuning with the classification loss, what about using both losses for fine-tuning? It was mentioned in Line 95 that "they are also mostly unable to improve upon state-of-the-art supervised methods when finetuned with full supervision", but there's no evidence to support this.


4. Since the paper is proposing to learning compositionality with part-whole hierarchy, a more convincing experiment would be running with a dataset that contains part-whole hierarchy (e.g. PartNet [a]), and provide a quantitative analysis on the learnt hierarchy v.s. other baselines that learns part whole hierarchy.


5. (minor point) In the Fig. 4 visualization, it seems the clustering is not great? If we only focus on the light green color, they are spread in many regions in the graph, does this mean the clustering of light green class is not great?

[a] PartNet: A Large-scale Benchmark for Fine-grained and Hierarchical Part-level 3D Object Understanding
Kaichun Mo, Shilin Zhu, Angel X. Chang3, Li Yi, Subarna Tripathi, Leonidas J. Guibas, Hao Su

[b]How to represent part-whole hierarchies in a neural network
Geoffrey Hinton

---

> ### Author Response · Authors · 2022-07-29
> **Response to reviewer r1Np**
>
> We thank the reviewer for their interest in our work. In the following the address the raised concerns on a point by point basis.
>
> > 1.“The definition of part-whole hierarchy seems controversial. In the paper, the part is defined as the ancestor of the whole object, and different object can share same ancestor...”
>
> We agree with the reviewer that our definition of hierarchy might appear unusual but it is well-motivated by considering the properties of the hyperbolic space. It is also tied to the specific problem we address in this paper, i.e., whole point cloud classification rather than part segmentation, for which, in fact, the optimal hierarchy in the hyperbolic space is the one mentioned by the reviewer and subject of our future work. To see this we report part of the response to reviewer 45jd. First, we notice that the Poincarè Ball model of hyperbolic space owes its effectiveness in our scheme to the exponential growth of the space volume as a function of the radius. This means that embeddings of things that are used to directly estimate the task labels should be placed close to the boundary of the Ball, as this is where they will be able to cluster better. In the case of whole point cloud classification, our estimated label is derived from the embedding of the entire point cloud, so this is what we aim at placing on the boundary. At the same time, the working of the hyperbolic space when embedding a hierarchy, is such that elements at higher levels of the hierarchy sit closer to the Ball center so that the geodesic distance approximates the tree-distance. This is why for the classification task considered in this paper, the natural hierarchy to consider is that of proto-parts with common shapes serving as ancestors of multiple objects. For part segmentation, the ideal hierarchy is flipped, as we desire the part embeddings to sit close to the Ball boundary and whole objects serving as common ancestors of their parts.
> In an ablation study that we did not report for reasons of brevity, we considered the flipped hierarchy suggested by the reviewer which resulted in poor performance. Indeed, this seems to go against the natural tendency of the model to follow the hierarchy we discussed in the paper.
>
> > 2. “ Overclaiming. I don't agree with the author the paper is learning to "promotes the part-whole hierarchy of compositionality in the hyperbolic space" Specifically, the only thing the paper is proposing a regularization on the feature space, therefore how the object is composed with different part is not clear, there's no explicit way of representing object as a hierarchical tree using the proposed method, not mentioning the compositionality. Besides, the way the paper is defining "parts" is through subsampling a small local region of the object (Line 188), this is not a semantic part and is only a subgroup of the object, and also the definition of part-whole hierarchy is controversial as above"
>
> In this work we augment point cloud classification architectures by proposing the hyperbolic embedding and a regularization specific to this space. Through the regularization, we impose that, as parts get larger, they become more specific to the corresponding object, while general parts can connect different classes of object. We believe that part size can be a reasonable proxy for specificity. We agree with the reviewer that there is not an explicit reasoning on compositionality, However, if we think of starting with a small part and then progressively adding points, we are moving from the center of the Poincarè Ball towards the boundary. This is what creates the property for which tracing the geodesic between two parts (or objects) at similar radius, results in passing through smaller parts that share a commonality of shape with the endpoints.
> We also agree that we are not explicitly interested in semantically meaningful parts. A “part” as considered by our own framework does not need to be its own semantic entity in order to provide the desired effect. In other words, a semantically well-defined leg is not necessarily more useful than a blob of points capturing a piece of leg and a piece of plane which may occur in chairs, tables, etc.. The lack of need for semantically-defined parts is also a feature of the method, as it limits labeling requirements. We are in fact ultimately interested in whole point cloud classification, not part segmentation, nor generative part-based modeling. Nevertheless, it could be interesting to explore if including semantic part labels in the definition of parts has an effect on performance.

---

> > ### Author Response · Authors · 2022-07-29
> > **Response - part 2**
> >
> > > 3. “The comparison is not a fair comparison. I appreciate the paper compared with multiple network architectures and see the improvement over compared baselines. However, since the main idea is proposing a regularization loss in the hyperbolic space, a fair comparison should be apply the same regularization loss, but in the Euclidean space. The compared results is only using supervised loss for pretraining and fine-tuning with the classification loss, what about using both losses for fine-tuning? It was mentioned in Line 95 that "they are also mostly unable to improve upon state-of-the-art supervised methods when finetuned with full supervision", but there's no evidence to support this. "
> >
> > The compared results is only using supervised loss for pretraining and fine-tuning with the classification loss, what about using both losses for fine-tuning? It was mentioned in Line 95 that "they are also mostly unable to improve upon state-of-the-art supervised methods when finetuned with full supervision", but there's no evidence to support this.  ”
> > We agree with the reviewer that it is important to test the effectiveness of the regularization loss in the Euclidean space. This is done in Table 3 where EuCoRe is the same regularization loss but in the Euclidean space. As it can be noticed, the Euclidean space provides only a small improvement.
> > Regarding the comparisons with other models, we consider methods as they have been presented in the literature. Several of the techniques we report in the “finetuned” category have been proposed in the context of self-supervised learning and exhibit strong performance in that setting. However, once labels are added and the self-supervised pretraining is finetuned, they do not significantly outperform their counterparts trained without the self-supervised schemes. This is the claim we make at Line 95, and it is supported by the results in Table 1. Specifically, we can see that Self-Recon [33], STRL [34], DCGLR [7] are unable to improve the performance of DGCNN by more than 0.3 OA points and PointGLR [5] only improves PointNet++ by 0.1 points. These results are reported by the original authors themselves in the usual pretraining followed by finetuning configuration, and we assumed that if their methods worked better by using both losses for finetuning as suggested by the reviewer, the original authors would have reported it.
> >
> > > 4. “Since the paper is proposing to learning compositionality with part-whole hierarchy, a more convincing experiment would be running with a dataset that contains part-whole hierarchy (e.g. PartNet [a]), and provide a quantitative analysis on the learnt hierarchy v.s. other baselines that learns part whole hierarchy.”
> >
> > We would like to remark that our goal is not to learn part-whole hierarchy in the semantic sense that a generative model or a part segmentation model would do. Rather, we want to exploit   a hierarchy between whole objects and some kind of partial shapes that allows to induce better clusters for classification of the objects. In fact, the method does not provide any kind of part segmentation output.
> > We performed an experiment that follows the spirit of the reviewer’s suggestion to highlight the properties of our regularized space. We consider the embeddings of semantic parts, as extracted from ShapeNet Parts using the part labels. We progressively combine semantic parts, embed them and check where they land in the hyperbolic space by measuring the hyperbolic norm of the embedding (distance from Ball center). This is presented in the following table. We can see that as parts get progressively combined to form the whole object, the norm increases, thus moving the embedding from the center of the ball towards the edges in the typical fashion of embedded hierarchies.
> >
> > Legend: Part or composition of parts (hyperbolic norm)
> >
> > Table (5.32)
> > Plane+uprights (4.56)
> > Legs+uprights (2.08)
> > Plane (4.07)
> > Legs (2.05)
> > Uprights (1.99)
> >
> > Aircraft (4.98)
> > Wings+tail+engines (4.56)
> > Wings+tail (4.45)
> > Fuselage+tail (4.23)
> > Engines (4.44)
> > Wings (4.22)
> > Fuselage (3.37)
> > Tail (2.94)
> >
> >
> > > 5. (minor point) In the Fig. 4 visualization, it seems the clustering is not great?
> >
> > Figure 4 is a qualitative 2-d visualization of a 256-d dimensional hyperbolic feature space. These visualization techniques can be quite unstable due to the aggressive dimensionality reduction in the hyperbolic space, so they must be used only for a high-level qualitative overview. In this case, it is mostly useful to see that larger dots are closer to the boundary, that there is some clustering and that some small dots seem to bridge multiple large dots clusters.

---

> > > ### Comment · Reviewer_r1Np · 2022-08-04
> > > **Thanks for the responses**
> > >
> > > Thank the author for providing a detailed rebuttal!
> > >
> > > I still have some questions about the rebuttal.
> > >
> > > 1. In Table 3, comparing with DGCNN-EuCoRe and DGCNN-HyCoRe, the improvement is only ~1% in average accuracy improvement across different dimensions. DGCNN-EuCoRe is the direct baseline for this paper, however, this improvement is not huge in the ScanObjectNN dataset.
> > >
> > > 2. It's mentioned that
> > > > "we want to exploit a hierarchy between whole objects and some kind of partial shapes that allows to induce better clusters for classification of the objects. In fact, the method does not provide any kind of part segmentation output."
> > >
> > > > " This is what creates the property for which tracing the geodesic between two parts (or objects) at similar radius, results in passing through smaller parts that share a commonality of shape with the endpoints.".
> > >
> > > The major problem I see in this way is that the "part" defined in this paper only means spatially connected subgroups of the whole shape, and achieving this kind of compositionality is quite straightforward, e.g. sort the points along one axis, and gradually increase along this axis. It's also agreed with the authors that there's no explicit compositionality in the formulation. I don't think it is reasonable to claim the compositionality in the paper.

---

> > > > ### Author Response · Authors · 2022-08-05
> > > > **Further clarifications**
> > > >
> > > > Regarding the experiment in Table 3, we would like to remark that EuCoRe is also a novel regularization method for the classification problem based on the ideas presented in this paper. Table 3 just shows that implementing those ideas in the hyperbolic space is much more effective. We thus believe that the correct baseline is the published literature presented in the other Tables, more than a suboptimal implementation of our own contribution. Nevertheless, a 1 point improvement over our own EuCoRe is very significant on the difficult ScanObjectNN dataset, where improvements by state-of-the-art methods are often even smaller.
> > > >
> > > >
> > > > We agree with the reviewer that explicit compositionality in the sense of hierarchical structural inference or generative modeling is not our focus. We are ultimately interested in boosting the performance of a classifier via regularization, and it seems from experimental evidence that the "straighforward kind of compositionality", as described by the reviewer, we exploit is quite effective and, in fact, more effective than what has been studied in the literature and reported in our experiments tables. We would be glad if our paper pointed out this important phenomenon and further work focused on integrating more refined kinds of compositional models.

---

### Official Review · Reviewer_45jd · 2022-07-21

**Rating:** 7
**Confidence:** 4
**Soundness:** 4 excellent
**Presentation:** 4 excellent
**Contribution:** 4 excellent

**Summary:**

The paper proposes learning shape representation for the 3D shape classification task,
by regularizing embeddings in hyperbolic space. The intuition on which this paper is based
is that complex shapes can be made by combining simpler parts and this composition can
be explained by a tree-like hierarchy. The paper proposes regularizing shape embeddings
such that simplest and most basic parts are embedded at the root level and entire shapes
are embedded at the leaf level, where embeddings are defined using hyperbolic space.
Specifically, a shape embedding is first mapped to hyperbolic space and then a mobius layer
is applied to project it to Poincare ball. The paper proposes two regularization, first is to
encourage the whole shape embeddings to be close to leaf level and part level embeddings
are close to the root. The second regularization encourages a shape and its parts
to be closed to each other in embedding space and far from embeddings of parts from different
shapes.
This approach consistently improve the performance of shape classification task on several
neural network architectures.

**Questions:**

1. Is it easy to extend this approach to semantic segmentation task? If yes, why did authors not include those experiments? If not, can authors include some discussion on this matter.

**Limitations:**

Yes.

**Strengths And Weaknesses:**

**Strengths**
1. The paper is clearly written and explains the motivation of the proposed approach well.
2. The proposed approach is well described
3. All experiment details are provided ensuring reproducibility.
4. The proposed approach consistently improve the performance using several neural network architecture, ensuring generalizability.
5. All ablations are provided that all components are pertinent.
**Weakness**
1. The main weakness I see is that only shape classification is chosen to benchmark the approach. Specifically, since the embeddings are better aware of the composition the embeddings can shine in part segmentation task.

---

> ### Author Response · Authors · 2022-07-29
> **Response to Reviewer 45jd**
>
> > Is it easy to extend this approach to semantic segmentation task? If yes, why did authors not include those experiments? If not, can authors include some discussion on this matter.
>
> We thank the reviewer for their positive comments on our work. We agree with the reviewer that segmentation would be an interesting task to consider and, in fact, we are actively working on it. However, we have found that there are substantial differences between the optimal way to regularize the tasks of whole point cloud classification and part segmentation. This deserves a rather lengthy discussion and analysis which cannot fit a single paper, so we decided to only focus on classification for this work.
>
> The reason why regularizing part segmentation is different from regularizing classification is found in the need to impose opposite hierarchies in the hyperbolic space. This also addresses the comments from other reviewers observing our unusual definition of hierarchy. First, we notice that the Poincarè Ball model of hyperbolic space owes its effectiveness in our scheme to the exponential growth of the space volume as a function of the radius. This means that embeddings of things that are used to directly estimate the task labels should be placed close to the boundary of the Ball, as this is where they will be able to cluster better. In the case of whole point cloud classification, our estimated label is derived from the embedding of the entire point cloud, so this is what we aim at placing on the boundary. At the same time, the working of the hyperbolic space when embedding a hierarchy, is such that elements at higher levels of the hierarchy sit closer to the Ball center so that the geodesic distance approximates the tree-distance. This is why for the classification task considered in this paper, the natural hierarchy to consider is that of proto-parts with common shapes serving as ancestors of multiple objects. For part segmentation, the ideal hierarchy is flipped, as we desire the part embeddings to sit close to the Ball boundary and whole objects serving as common ancestors of their parts. Indeed, in our experiments we observed that these two hierarchies emerge naturally just by moving to the hyperbolic space without any regularizers forcing them (albeit they are very rough without explicit regularization). Our preliminary results on part segmentation suggest that substantial gains are also possible on that task by following its optimal hierarchy. However, as mentioned, this will be subject of future work. We will briefly summarize this discussion in the conclusions section of the final version of the paper.

---

### Meta-Review · Area_Chair_EbLP · 2022-08-24

**Recommendation:** Accept
**Confidence:** Certain

**Metareview:**

The paper presents a regularization for point cloud representation learning aiming to promote a part-whole hierarchy through a hyperbolic space. Most of the reviewers agree the idea of using the hyperbolic space is new and interesting. The experiment results seem to be sufficient. There was some confusion on how part is defined and compositionality in the paper. But the AC feels the paper has sufficient merit to be published It is required that the authors incorporate the reviewer feedbacks in the revised manuscripts.

**Award:**

No

---

### Decision · Program_Chairs · 2022-09-14

Accept